# Precision Oncology Framework Using Circulating Tumor Cells

**DOI:** 10.3390/ijms26125539

**Published:** 2025-06-10

**Authors:** Fumihiko Kakizaki, Kyoichi Oshiro, Yuya Enoki, Kana Kawanishi, Norikazu Masuda, Hisatsugu Maekawa, Jun Matsubayashi, Masahiro Kawashima, Hiroyuki Miyoshi, Yukitoshi Takemura, Kazutaka Obama

**Affiliations:** 1Department of Personalized Cancer Medicine, Graduate School of Medicine, Kyoto University, 53 Shogoin-Kawahara-cho, Sakyo-Ku, Kyoto 606-8507, Japan; 2AFI Corporation, Medical Innovation Center, Building, Kyoto University, 53 Shogoin-Kawahara-cho, Sakyo-Ku, Kyoto 606-8507, Japan; 3Department of Surgery, Graduate School of Medicine, Kyoto University, 54 Shogoin-Kawahara-cho, Sakyo-Ku, Kyoto 606-8507, Japan; 4Department of Breast Surgery, Graduate School of Medicine, Kyoto University, 54 Shogoin-Kawahara-cho, Sakyo-Ku, Kyoto 606-8507, Japan

**Keywords:** CTC, colorectal cancer, breast cancer, precision oncology, ctDNA

## Abstract

Circulating tumor cells (CTCs) are multifaceted biomarkers with significant potential for precision oncology, offering opportunities to refine diagnoses and personalize treatments across various cancer types, including colorectal and breast cancer. CTC assays serve as reliable prognostic indicators, even during chemotherapy and/or molecularly targeted therapies. Notably, CTCs exhibit heterogeneity that gradually develops during carcinogenesis and becomes more pronounced in advanced disease stages. These intra- and intertumoral heterogeneities pose challenges, particularly when drug-resistant clones emerge following therapy. The dynamic behavior of CTCs provides valuable insights into treatment response and prognosis. Extensive efforts have led to the development of technologies for effective CTC isolation, accelerating their clinical implementation. While both CTC and circulating tumor DNA (ctDNA) tests offer prognostic value, they reflect different aspects of tumor biology: CTC counts indicate tumor progression, while ctDNA levels correlate with tumor burden. The combined analysis is expected to yield complementary insights. CTC tests are feasible in general hospitals and may serve as tumor markers comparable to, or even superior to, conventional markers such as carcinoembryonic antigen (CEA) and carbohydrate antigen 19-9 (CA19-9) for colorectal cancer, and CA15-3 for breast cancer. Early incorporation of CTC tests into routine blood panels appears to be a rational and promising approach.

## 1. Introduction

Precision oncology is an approach to cancer treatment that integrates the molecular and cellular features of tumors, the characteristics of the tumor microenvironment, and patient-specific factors to develop individualized therapeutic strategies [1,2,3]. Recent clinical trials of molecularly targeted therapies have demonstrated significant improvements in survival outcomes among patients with cancers, including colorectal and breast carcinomas, whose tumors harbor specific molecular alterations. Currently, such biomarker-driven therapies are available for an increasing number of cancer types [3].

Circulating tumor cells (CTCs) are multifaceted biomarkers that hold significant promise for the precision oncology, offering the potential to refine diagnoses and tailor treatments. One key advantage of CTCs is their minimally invasive nature, particularly compared to traditional biopsies, as they are obtained solely through blood collection. Once isolated, CTCs can undergo various types of genomic and molecular analyses and are often referred to as a real-time “liquid biopsy” for cancer patients [4,5,6].

CTC-derived data have applications in several clinical settings, including tumor screening and early diagnosis, detection of recurrence, and surveillance of disease progression [5,7]. The genomic profiling of individual CTCs is expected to be a critical component of precision oncology and is predicted to enhance the precision of cancer treatments for approximately 20% of molecularly matched patients [1,8]. Numerous basic and clinical studies have demonstrated that CTC enumeration serves as a predictive and prognostic biomarker [7].

In addition to CTCs, another blood-based biomarker is circulating tumor DNA (ctDNA). ctDNA, released from apoptotic or necrotic tumor cells, serves as a biomarker for detecting somatic variants present in tumor tissue [9]. Like CTCs, ctDNA can be obtained through a simple blood draw, offering a minimally invasive alternative to traditional biopsies. Although ctDNA cannot be used for transcriptional or pathological analyses, it has several clinical applications, including early cancer detection, detection of minimal/molecular residual disease (MRD), and monitoring of treatment response [9].

In this review, we outline the clinical applications of CTCs in relation to metastasis, molecular pathology, prognostic biomarkers, comparison with ctDNA, and detection methods. We place particular emphasis on colorectal cancer, which typically exhibits lower CTC levels, and breast cancer, which tends to show higher CTC levels. Globally, colorectal and breast cancers are the second and fourth leading causes of cancer-related mortality, and the third and second most commonly diagnosed malignancies worldwide, respectively [10]. These cancers have been extensively studied in the context of CTC analysis due to their clinical relevance, distinct patterns of metastasis, and the availability of comprehensive data on CTC isolation. Including both tumor types enables comparative evaluation of CTC characteristics across malignancies with significant clinical impact.

## 2. CTCs in the Metastatic Process

Carcinomas invade surrounding tissues and enter the bloodstream via nearby veins. Portions of the primary tumor dissociates, circulating as single cells or clusters within blood vessels before being transported to distant organs. These CTCs can adhere to the vessel walls, extravasate, and, albeit at a low frequency, establish metastatic colonies (Figure 1) [11].

In the bloodstream, colon cancer cells are frequently transported to the liver via the mesenteric and portal veins and subsequently reach the lungs (Figure 1a) [12,13]. In contrast, lower rectal cancer cells have a greater tendency to metastasize to the lungs directly via the inferior rectal vein, inferior vena cava, and heart (Figure 1a) [12,13]. Similarly, breast cancer cells are thought to spread hematogenously to the bones, lungs, and liver via the venous circulation and heart (Figure 1b), and colonize distant organs with molecular signature-driven organotropism [14].

Although cancer researchers have studied the metastatic process for over 120 years, it is only recently that molecular insights have emerged clarifying the organotropism of certain tumor types, including colorectal and breast cancer (Figure 1) [15,16]. Namely, cancer cell-secreted factors, including tumor-derived extracellular vesicles and particles (EVPs), circulate and act on distant organs to establish a tumor microenvironment (TME) that is favorable for the initiation of metastasis. These changes lead to the formation of “pre-metastatic niches” (PMNs), which are receptive to the colonization of CTCs [17]. In addition, tumor-associated macrophages (TAMs) are the particularly abundant stromal cells in TME of most solid tumors, and a high TAM density is associated with poor prognosis in cancer patients [16,18].

Nevertheless, cancer metastasis is considered an inefficient process [19]. CTCs are subjected to various forms of cell death due to factors such as detachment-induced apoptosis (anoikis), immune attacks by CD8⁺ T cells and natural killer (NK) cells [16], and fluid shear stress as they traverse capillaries [20]. CD8⁺ T cells and NK cells typically induce apoptosis in CTCs through the secretion of perforin and granzymes, as well as through the expression of death ligands such as Fas ligand (FASL) and tumor necrosis factor-related apoptosis-inducing ligand (TRAIL) [21]. On the other hand, neutrophils contribute to both tumoricidal and tumor-promoting phenotypes [16,22,23].

It is well established that CTCs undergoing epithelial–mesenchymal transition (EMT) express mesenchymal markers. Moreover, some CTCs have been reported to maintain a quiescent or dormant state [24]. These observations suggest that the behavior of CTCs reflects not only their ability to adapt to the circulatory environment but also their inherent plasticity.

Several studies suggest that CTC clusters exhibit higher metastatic potential than single CTCs [25,26,27], but due to their rarity, reports on their clinical significance remain limited [7]. CTCs are also believed to reflect the biology of metastasis itself [28]. Therefore, methods that enable early detection of colorectal and breast cancer and real-time monitoring of CTCs may hold great promise for improving clinical outcomes and patient survival [29].

## 3. Heterogeneity and Molecular Pathology of CTCs

Tumor heterogeneity, a key factor in the emergence of drug-resistant cells following chemotherapy and/or endocrine therapy, can arise during the development and progression of primary tumors [30], or be induced by various treatments within these primary tumors [31,32]. Consequently, multiple tumor cell subtypes appear as CTC subpopulations in the later stages of the disease [33,34].

In primary colorectal tumors, driver mutations in genes such as *adenomatous polyposis coli* (*APC*), *tumor protein p53* (*TP53*), *Kirsten rat sarcoma viral oncogene homolog* (*KRAS*), and *SMAD family member 4* (*SMAD4*) are associated with multistep tumorigenesis through the adenoma–carcinoma sequence [30,35,36,37]. Mutations in *KRAS* at codon 12 or 13 (observed in 35% to 45% of cases) or the *v-raf murine sarcoma viral oncogene homolog B* (*BRAF*) V600E mutation (approximately 10%) confer resistance to cetuximab-based therapies in metastatic colorectal cancer (mCRC) [38]. Notably, *KRAS* and *BRAF* mutations are mutually exclusive in colorectal cancers [39].

Interestingly, several studies have highlighted inconsistencies between the molecular profiles of CTCs and their corresponding primary tumors. Mutation analyses of driver genes in colorectal cancer CTCs have revealed discordance between CTCs and primary tumors [40,41]. Specifically, mutations in *KRAS*, *BRAF*, or *phosphatidylinositol-4,5-bisphosphate 3-kinase catalytic subunit alpha* (*PIK3CA*) were detected in CTCs but were absent in the primary tumors [40,41], suggesting a potential role in resistance to molecularly targeted therapies [42,43].Similarly, in luminal-type breast cancer, certain activating mutations are detected specifically in CTCs but are absent in the primary tumor. For instance, mutations in *estrogen receptor 1* (*ESR1*) and *phosphatidylinositol-4-phosphate 3-kinase catalytic subunit type 2 gamma* (*PIK3C2G*) have been identified in metastatic lesions but occur at low frequencies in primary tumors [44,45]. Such discrepancies may reflect tumor evolution under therapeutic pressure and the emergence of resistant clones [46,47]. On the other hand, other reports have shown little or no difference in driver gene mutations between the primary tumors and metastatic lesions [30,48].

The heterogeneity of CTCs can be demonstrated by their expression of key molecular markers. In metastatic colorectal cancer, this heterogeneity can be detected through the distinct expression patterns of *caudal type homeobox 1* (*CDX1*), *CDX2*, and/or *catenin beta 1* (*CTNNB1*) (Figure 2a). In metastatic breast cancer, it is reflected in the varied expression patterns of *ESR1*, *ESR2*, *progesterone receptor* (*PGR*), and/or *erb-b2 receptor tyrosine kinase 2* (*ERBB2*), consistent with the heterogeneity observed in the primary tumors and/or metastatic lesions (Figure 2b) [25,49,50,51,52,53,54].

One of the major contributors to gene expression heterogeneity is EMT, a process that facilitates tumor progression and metastasis [31,55]. Since *vimentin*, a stromal intermediate filament protein is highly expressed during EMT, antibodies against cell-surface vimentin (CSV), a cancer-specific antigen, have been used to detect live CTCs in patients with EMT-associated carcinoma or malignant sarcomas [56,57,58,59]. Notably, approximately 70% of CTCs in both metastatic colorectal and breast cancers express the epithelial cell adhesion molecule (*EPCAM*), members of the Keratin (*KRT*) family, and/or vimentin (*VIM*) (Figure 2a,b).

A subset of CTCs in recurrent or metastatic breast cancer exhibits a KI67-negative state, indicative of quiescence or dormancy, during which the cells undergo cell cycle arrest [60,61] (Figure 2b). The expression levels of proliferation markers such as *marker of proliferation Ki-67* (*MKI67*), *telomerase reverse transcriptase* (*TERT*), myelocytomatosis oncogene (*MYC*), and *cyclin B1* (*CCNB1*) are significantly downregulated in CTCs compared to those in their parental cell lines or in primary tumors (Figure 2b) [62]. Importantly, this quiescent phase contributes to resistance to anticancer drugs and facilitates tumor recurrence.

These genomic, transcriptomic, and phenotypic studies demonstrate that CTCs can exhibit significant heterogeneity, even within the same patient. As a result, identifying reliable biomarkers for CTC detection remains challenging [63]. Consequently, researchers have explored alternative approaches for CTC detection, which are discussed in Section 6.

**Figure 2 ijms-26-05539-f002:**
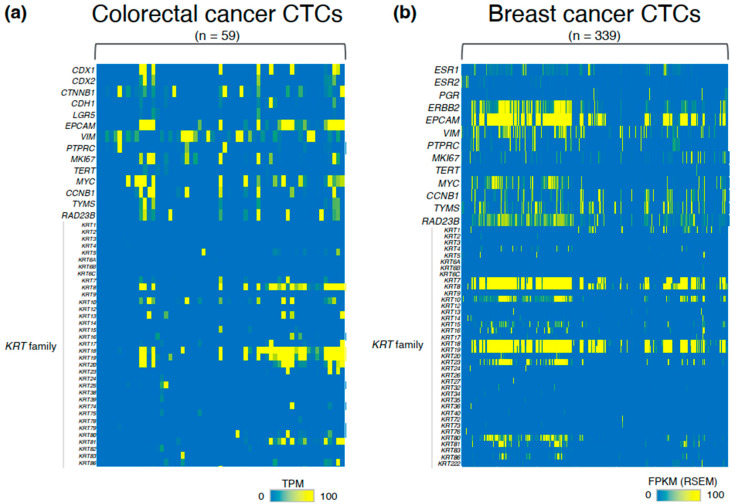
Heterogeneity in CTC transcriptomes of metastatic colorectal and breast cancers. (**a**,**b**) Heatmaps showing CTC transcriptomic profiles in metastatic colorectal (n = 59; panel (**a**)) and breast cancers (n = 339; panel (**b**)) [25,49,50,51,52,53,54,64]. In panel (**b**), ctcRbase integrates seven datasets (GSE111065, GSE51827, GSE55807, GSE67939, GSE75367, GSE86978, and GSE41245) as previously described [64]. Panels (**a**) and (**b**) present expression values in transcripts per million (TPM) and fragments per kilobase of transcript per million mapped reads (FPKM), respectively. These heatmaps were reconstructed by the authors based on data reported in previous studies [25,49,50,51,52,53,54,64]. As only limited genomic data on CTCs was available in these datasets, gene mutation data is not presented.

## 4. CTCs as Prognostic and Predictive Biomarkers in Cancer Therapy

Based on accumulated evidence highlighting the prognostic significance of CTCs in various cancers, researchers have investigated the clinical utility of CTC enumeration. The dynamics of CTCs provide insights into therapeutic response levels. In colorectal cancer patients, the presence of CTCs has been significantly correlated with poor overall survival (OS), poor disease-free survival (DFS), poor progression-free survival (PFS), and advanced tumor, node, and metastasis (TNM) stages [65,66,67,68]. 

In a meta-analysis of colorectal cancer patients with mixed stages, CTC counts of ≥1, ≥2, and ≥3 were associated with hazard ratios (HRs) for overall mortality (95% CI) of 2.33 (0.44–12.18), 2.12 (1.13–3.98), and 1.72 (1.28–2.32), respectively [29]. A threshold of ≥3 CTCs per 7.5 mL of blood has frequently been used as a marker to define “high CTC” status in metastatic colorectal cancer [4,29,69,70,71,72]. Additionally, CTC enumeration demonstrated the added benefits of bevacizumab plus FOLFOXIRI (folinic acid, fluorouracil, oxaliplatin, and irinotecan) compared to bevacizumab plus FOLFOX (folinic acid, fluorouracil, and oxaliplatin) in colorectal cancer patients [4,72,73].

Interestingly, CTC-based stratification identified two distinct response groups among metastatic colorectal cancer patients treated with bevacizumab and FOLFOX: a favorable response group (complete response [CR], partial response [PR], or stable disease [SD]) and an unfavorable response group (progressive disease [PD]) [74]. Separately, in locally advanced rectal cancer, CTCs expressing thymidylate synthase (TYMS) and RAD23 homolog B (RAD23B) were identified as markers of resistance to chemotherapy and radiotherapy (Figure 2) [75].

In clinical trials involving early- and late-stage breast cancer patients undergoing chemotherapy, CTC positivity was associated with unfavorable OS and DFS [76,77,78,79]. A meta-analysis pooling data from 21 studies, which included 2,239 non-metastatic breast cancer patients who received neoadjuvant chemotherapy, demonstrated that higher CTC counts correlated with poor survival outcomes [80]. Specifically, the hazard ratios (HRs) for death (95% CI) associated with CTC counts of 1, 2, 3–4, and ≥5 were 1.09 (0.65–1.69), 2.63 (1.42–4.54), 3.83 (2.08–6.66), and 6.25 (4.34–9.09), respectively.

Furthermore, CTC counts serve as prognostic markers in metastatic breast cancer, not only in patients receiving chemotherapy but also in those undergoing molecular-targeted therapies, such as letrozole plus bevacizumab and anti-HER2 (human epidermal growth factor receptor 2, also known as ERBB2) monoclonal antibodies [81,82,83,84]. Changes in CTC counts during treatment are significantly associated with PFS and OS, underscoring the value of CTC tracking in improving prognostic accuracy for metastatic breast cancer [7,79,85,86].

Therefore, CTC monitoring can serve as a complementary tool for therapeutic decision-making, clinical evaluation, and treatment efficacy assessment. Enumeration of CTCs provides real-time indicators of tumor response and has been recognized as an effective prognostic biomarker. In patients with early-stage, operable, estrogen receptor-positive (ER+) breast cancer, CTC counts have also been used as indicators for detecting minimal/molecular residual disease (MRD) up to four years earlier [87]. However, combining CTCs with ctDNA may improve the sensitivity of MRD detection [88], as discussed below.

## 5. Feasibility of the CTC Test: A Comparison with the ctDNA Test

CTCs exhibit a variety of features over time in individual patients, with their numbers typically increasing prior to recurrence [87,89]. While both CTCs and ctDNA offer prognostic information, CTC count reflects cancer progression, whereas ctDNA quantity provides an indication of tumor burden [7,89]. Although few colorectal cancer studies have reported a direct comparison between CTC and ctDNA detection [90], the breast cancer study shows concordance between them was approximately 48% in early-stage breast cancer and 67% in the metastatic setting, with the latter showing a higher level of concordance [91]. These findings suggest that CTC and ctDNA detection provide complementary insight. The CTC test is superior to the ctDNA test for downstream analyses such as cytological tests, chemosensitivity tests, and gene expression profiling (Figure 2) (Table 1). However, the ctDNA shows strengths in sequential quantification of tumor burden [92,93]. Regarding the ctDNA test, it is important to consider that a laboratory testing consortium has highlighted the following limitations: “ctDNA exists in extremely low quantities and is highly fragmented, requiring deep sequencing of a large number of DNA fragments. Additionally, a bioinformatics pipeline capable of detecting variants with low allele frequencies is necessary... [94]” This suggests that the ctDNA test may not be feasible in general hospitals (Table 1). In contrast, CTCs can be analyzed if the appropriate equipment is available, and many detection systems are compatible with pathology laboratories to a certain degree [95,96,97]. Once CTCs are isolated using minimally damaging separation systems such as ClearCell FX1, CROSSORTER, or Parsortix (see Section 7) [98,99,100], they can be processed using standard cytological procedures. Specifically, the cells are attached to glass slides via routine cytocentrifugation, stained using conventional immunocytochemistry, and examined under a standard light microscope. For enhanced sensitivity, quantitative polymerase chain reaction (qPCR) can also be performed using Real-Time PCR instruments that are widely available in clinical laboratories (Table 1) [101].

## 6. CTC Separation Techniques

CellSearch is a CTC detection system that was approved by The United States Food and Drug Administration (FDA) in 2004 for use in the diagnosis and prognosis prediction of metastatic colorectal, breast, and prostate cancer [97]. In addition, Parsortix was also granted de novo classification by the FDA in 2022 for isolation of CTC in metastatic breast cancer [98]. Furthermore, various researchers and developers have meticulously designed devices for CTC isolation. The following sections and Table 2 describe these techniques in more detail.

### 6.1. Size-Based Separation, Including Hydrodynamic Methods

Size-based separation utilizes the fact that CTCs are generally larger than normal blood cells. By employing fine filters of specific sizes, CTCs can be selectively separated. For instance, white blood cells and red blood cells that are smaller than 8 µm can pass through the filter, while CTCs, which typically range from 10 to 20 µm, are retained and collected. A representative system using this principle is Isolation by Size of Epithelial Tumor Cells (ISET) [106].

Another approach in this category involves microfluidic constrictions, where microfluidic chips are designed with narrow constriction regions. Owing to their larger size, CTCs are physically unable to pass through these narrow constrictions, enabling their separation and collection. This method has the added advantage of reducing the risk of filter clogging, a common issue in membrane filtration systems.

Hydrodynamic separation techniques address similar challenges by exploiting fluid dynamics for CTC isolation [107]. One such method is inertial microfluidics, which utilizes the inertial forces and dean flow characteristics generated as the fluid passes through microchannels. Larger CTCs are directed toward the center or specific regions of the flow path, allowing for their separation from smaller blood cells. Spiral microfluidics and Dean Flow Fractionation are well-known systems utilizing this approach [108,109].

Another hydrodynamic separation method is deterministic lateral displacement (DLD) [109], which employs microfluidic channels with systematically arranged obstacles. As cells pass through these obstacles, they follow different trajectories based on their sizes, enabling effective separation. This technique is advantageous for high-throughput separation processes.

### 6.2. Density-Based Separation

Density-based separation relies on the differences in density between CTCs and other blood cells, such as red and white blood cells. Centrifugal force is applied to separate cells based on their density differences. Common techniques include density gradient centrifugation using Ficoll or OncoQuick to concentrate CTCs [110,111]. Additionally, some microfluidic chips integrate centrifugal separation functions to enhance this approach [112].

### 6.3. Immunoseparation

Immunoseparation methods leverage the expression of specific surface markers on CTCs for selective isolation [106]. Positive selection involves capturing CTCs using antibodies targeting specific markers such as epithelial cell adhesion molecule (EpCAM). Systems such as CTC-iChip [107], which combines immunomagnetic separation with fluidic separation, and CTC-Chip [113], which captures CTCs using antibody-coated micropillars, are based on this principle. Negative selection, on the other hand, removes normal blood cells using antibodies targeting markers like CD45 and CD34. By eliminating these cells, only CTCs remain, including those that may be EpCAM-negative. This method enables the isolation of a broader spectrum of CTC subpopulations.

Immunomagnetic beads, such as those coated with EpCAM antibodies, allow for CTCs to be separated under a magnetic field. Representative systems include CellSearch [97], an FDA-approved CTC isolation platform, and MagSweeper [62], which integrates microfluidic chips with magnetic separation capabilities.

### 6.4. Acoustic Separation

Acoustic separation employs standing ultrasonic waves to manipulate cells based on their size, density, and compressibility [114]. Since CTCs and normal blood cells exhibit different physical properties, ultrasound waves can be used to selectively direct CTCs into specific regions for collection. Acoustophoresis, also known as acoustic microfluidics, is a well-established system based on this principle.

### 6.5. Dielectrophoresis (DEP)

DEP is a separation technique that utilizes non-uniform electric fields to induce motion in cells based on their dielectric properties [99,115,116]. Cells respond differently to the electric field, either being attracted to regions of high field intensity (positive DEP) or repelled toward low-intensity regions (negative DEP). This unique behavior allows label-free separation of CTCs without the need for antibodies or chemical markers.

DEP is particularly effective in isolating CTCs, as their dielectric properties differ from those of normal blood cells. By incorporating microelectrodes into microfluidic chips, real-time manipulation and separation of cells can be achieved. A notable DEP-based system is DEPArray, which enables single-cell-level CTC isolation using microelectrode technology [115]. Moreover, the CROSSORTER system combines DEP with size-based separation to achieve low-damage CTC isolation [99].

**Table 2 ijms-26-05539-t002:** Overview of CTC separation techniques and their characteristics.

Separation Technique	Separation Principle	Advantages	Representative Systems	References
Size-based (including hydrodynamic)	Filtration, microchannel constrictions, inertial microfluidics	Simple, high-throughput, label-free	ClearCell FX1, CROSSORTER, CTC-iChip ^1^, ISET, Parsortix, VTX-1	[98,99,100,106,107,109,117,118]
Density-based	Centrifugation, gradient	Simple, scalable	Ficoll, OncoQuick, RosetteSep	[110,111,119]
Immunoseparation	Positive/negative selection	High selectivity, specificity	AdnaTest ^2^, CellSearch ^2^, CTC-iChip, GEDI Chip, HB-chip, MagSweeper ^2^	[62,106,107,113,120,121,122]
Acoustic-based	Ultrasonic field	Non-invasive, label-free	AcouTrap ^3^	[114]
DEP-based	Dielectrophoresis	Label-free, precise control	Apostream, CROSSORTER ^4^, DEPArray, 3DEP	[99,115,116,122]

^1^ Uses immunolabeling; ^2^ Uses magnetic labeling; ^3^ Separation relies on size, density, and compressibility; ^4^ Also incorporates size-based separation.

## 7. Low-Damage CTC Isolation Systems

Systems that are considered to cause low damage should meet the following conditions: (1) minimal physical stress, avoiding strong centrifugal forces, shear forces, or pressure; (2) label-free separation, meaning no need for antibodies or magnetic beads, thereby preserving the integrity of the cell surface; and (3) the ability to maintain cells in a viable state. Several systems fulfill these criteria. The Parsortix PC1 System employs microfluidic technology to capture cells based on size and deformability, without requiring chemical treatments such as antibody labeling, allowing for the captured cells to be recovered alive (Table 2) [98]. It has been FDA-cleared and is well-suited for downstream analysis, including gene expression studies and cell culture. The ClearCell FX1 System utilizes inertial focusing technology to separate CTCs solely based on fluidic properties, minimizing shear stress and ensuring a high cell survival rate [100]. Since it does not require centrifugation, it enables label-free CTC isolation. The CROSSORTER integrates size-based and DEP-based technologies for label-free CTC separation (Table 2) [99]. Although it involves exposure to an electric field, it minimizes the need for physical and biological damage, thereby reducing cellular stress. This system has a high live-cell recovery rate, making it suitable for culture and genetic analysis. The VTX-1 Liquid Biopsy System is another label-free microfluidic platform that isolates CTCs based on size and morphology, without the use of antibodies, thereby reducing the risk of cell membrane damage [118]. These are particularly promising choices for researchers aiming to recover viable CTCs for subsequent culture or genetic analysis.

Further information on CTC isolation devices can be found in recent comprehensive reviews [7,104,123].

## 8. Current Challenges

While CTC-based technologies hold great promise for cancer diagnosis and treatment, their integration into routine clinical practice poses several clinical and technical challenges. The key points are as follows:(1)Definition of CTCs using both gene expression and mutation profiles.(2)Standardization of CTC isolation devices and protocols.(3)Formulation of guidelines by an international consortium.

To define CTCs, robust evidence is required to confirm that they are indeed cancer cells. Although this can be achieved by identifying driver-gene mutations, a number of studies fail to provide this essential evidence. Furthermore, the genetic heterogeneity of CTCs, which can vary across patients and even within the same patient over time, adds complexity to their definition and characterization.

Once such evidence is obtained, we will be able to establish benchmarks for CTC isolation devices and approve these devices for clinical use. These benchmarks will need to consider not only sensitivity and specificity but also the reproducibility of results across different laboratory settings and patient populations.

To accomplish this, it is crucial to collaborate with international CTC consortia. This collaboration will allow for the development of standardized guidelines, which will provide a unified approach to CTC isolation and clinical application. These guidelines will play a key role in ensuring that CTC-based technologies are implemented safely, effectively, and consistently across healthcare systems worldwide.

## 9. Precision Oncology Framework Using CTCs

After blood collection, CTCs are isolated from blood using low-damage separation systems such as ClearCell FX1, CROSSORTER, or Parsortix (Section 6 and Section 7). The isolated CTCs can then be processed using standard cytological techniques, including routine cytocentrifugation and immunocytochemistry. These cells can be examined and enumerated microscopically or alternatively detected via signal amplification by qPCR (Section 5). CTC profiling provides valuable insights into the biology of metastasis and contributes to clinical decision-making, particularly in the context of molecularly targeted therapies (Section 3 and Section 4).

CTCs may reflect metastatic potential, risk of recurrence, or prognostic relevance based on parameters such as cell count, the presence of gene mutations, or EMT characteristics (Section 2 and Section 3). Even if CTCs are not initially detectable at the start of treatment, they may emerge during long-term monitoring, sometimes years later (Section 4). Furthermore, various immune cell populations can influence the survival and proliferation of CTCs or disseminated tumor cells (DTCs), which may enter reversible states of dormancy or quiescence (Section 3).

Thus, CTCs represent multifaceted biomarkers with significant promise in precision oncology, offering the potential to enhance diagnostic accuracy and enable individualized treatment strategies. A key advantage of CTC analysis lies in its minimally invasive nature, requiring only blood collection compared to conventional tissue biopsies. CTC-based assays are expected to be integrated into clinical blood panels as tumor markers with greater clinical utility than traditional markers such as CEA, CA19-9, or CA15-3, thereby supporting oncologists in therapeutic decision-making.

Currently, over 3,000 clinical trials are investigating CTC-based diagnostics—approximately twice the number of trials focusing on ctDNA [124]. These ongoing studies have demonstrated the feasibility and clinical relevance of CTC testing in prospective trials involving patients with colorectal and breast cancers [125,126,127,128].

## 10. Limitations

The role of CTCs as a liquid biopsy has been recognized in many solid tumors; however, the prognostic implications of CTC counts and molecular characteristics are strongly influenced by both disease stage (early vs. metastatic) and tumor type. Therefore, combining distinct tumor types should be approached with caution, as it may lead to misinterpretation of the biological significance and clinical relevance of CTCs.

## Figures and Tables

**Figure 1 ijms-26-05539-f001:**
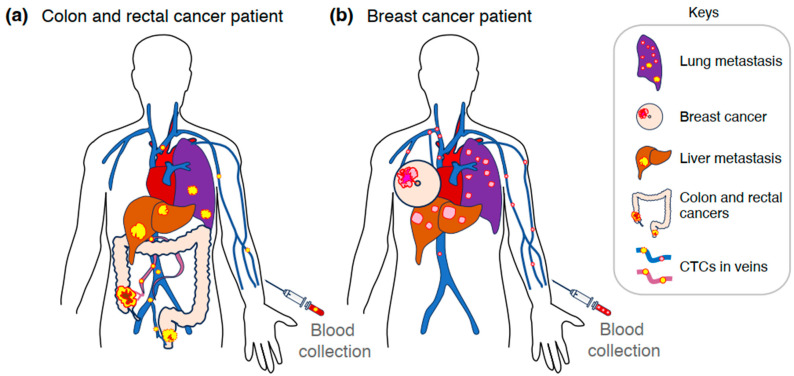
Overview of tumor dissemination and blood collection for CTC analysis. Colorectal (yellow, panel (**a**)) and breast (pink, panel (**b**)) carcinomas invade surrounding tissues, reach nearby veins (red purple, or blue), and enter the bloodstream as CTCs (yellow or pink circles). In panel (**a**), CTCs of colorectal cancer are frequently transported to the liver (brown) via the mesenteric and portal veins (red-purple) and subsequently reach the lungs (dark purple). In contrast, lower rectal cancer cells primarily metastasize to the lungs (dark purple) via the inferior rectal vein (blue), inferior vena cava, and heart. In panel (**b**), breast cancer cells are transported to the bones (not shown), lungs (dark purple), and liver (brown) through the venous circulation and heart.

**Table 1 ijms-26-05539-t001:** Comparison between CTCs and ctDNA.

Factor	CTCs	ctDNA
Discovery year	1869 ^1^	1977 ^2^
Detection target	Viable tumor cells	Cell-free DNA fragments
Release mechanism	Intravasation and circulation	Apoptosis or necrosis
Sensitivity	Varies by separation technique ^3^	Relatively high
Amplification in detection	Not required (but available) ^4^	Required
Monitoring MRD	Yes	Yes
Heterogeneity	Intra- and intertumor cells	Intra-/intertumor + mutated benign cells
Pathology/cytology	Yes	Not applicable
Functional assay	Yes	Not applicable
Gene expression	Yes	Not applicable
Gene alteration	Yes	Yes
Epigenetic changes	Yes	Yes (DNA methylation)
Characteristics	Preserved cell morphology and phenotype	DNA sequence and methylation
Chemosensitivity test	Yes	No
Feasibility in hospital setting	High (routine practice) ^5^	Limited (specialized setting) ^6^
Outsourced service	Available from commercial services	Available from commercial services

^1^, [102]; ^2^, [103]; ^3^, [104]; ^4^, [105]; ^5^, [95,96,97]; ^6^, [94].

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
