# Peer review of "Precision Oncology Framework Using Circulating Tumor Cells"

_ijms, 2025, doi:10.3390/ijms26125539_

Round 1
Reviewer 1 Report
Comments and Suggestions for Authors
The review by Kakizaki et al. focuses on the role of circulating tumor cells (CTCs) in colon and breast cancer, emphasizing their clinical applications, biological role in metastasis, comparison with circulating tumor DNA (ctDNA), and current detection approaches. While the role of CTCs as a liquid biopsy tool has been reported across several types of solid tumors, the prognostic significance of CTC enumeration and molecular characterization is highly dependent not only on the disease setting (early vs. metastatic) but also on the specific tumor type. Therefore, I do not find it appropriate to combine such distinct tumor types in a single review, as this approach may limit the overall scientific value and relevance of the article to the research community.
Author Response
Reviewer 1:
Comments and Suggestions for Authors
The review by Kakizaki et al. focuses on the role of circulating tumor cells (CTCs) in colon and breast cancer, emphasizing their clinical applications, biological role in metastasis, comparison with circulating tumor DNA (ctDNA), and current detection approaches.
→ We thank Reviewer 1 for the careful reading of our manuscript and the summary of its content.
Critique
While the role of CTCs as a liquid biopsy tool has been reported across several types of solid tumors, the prognostic significance of CTC enumeration and molecular characterization is highly dependent not only on the disease setting (early vs. metastatic) but also on the specific tumor type.
→ In response to Reviewer 1’s comment, we have clarified the disease settings and tumor types throughout the manuscript by adding the following phrases:
“In metastatic colorectal cancer” (P. 4 line 152)
“whereas in metastatic breast cancer” (P. 4 line 153)
“in both metastatic colorectal and breast cancer” (P. 4 line 162)
“in recurrent or metastatic breast cancer” (P. 4 line 165)
“in metastatic colorectal” (P. 5 line 176)
“in metastatic colorectal (n = 59; panel a)” (P. 5 line 177)
“colorectal cancer patients with mixed stages” (P. 5 line 191)
“early-stage operable ER(+) breast cancer,” (P. 6 line 221)
“metastatic colorectal, breast, and prostate cancer” (P. 7 line 257)
Therefore, I do not find it appropriate to combine such distinct tumor types in a single review, as this approach may limit the overall scientific value and relevance of the article to the research community.
→ To address this issue, we have added the following “Limitation” section.
“10. Limitations
The role of CTCs as a liquid biopsy has been recognized in many solid tumors; however, the prognostic implications of CTC counts and molecular characteristics are strongly influenced by both disease stage (early vs. metastatic) and tumor type. Therefore, combining distinct tumor types should be approached with caution, as it may lead to misinterpretation of the biological significance and clinical relevance of CTCs.” (P. 10 line 394)
→ In addition, to avoid any unintended misunderstanding, we have removed “in Colorectal and Breast Cancer Patients” from the paper title (P. 1 line 2) and revised related parts of the text accordingly.
“across various cancer types, including colorectal and breast cancer” (Abstract, P. 1 line 18)
“Carcinomas” (P. 2 line 74)
“primary tumors” (P. 4 line 126)
“various cancers” (P. 5 line 186)
→ Although we certainly understand the reviewer's concern regarding the inclusion of distinct tumor types in a single review, we would like to point out that several well-regarded reviews have addressed multiple cancer types together. For instance, Pantel et al. published a comprehensive review on CTCs in breast, colorectal, non-small-cell lung, and prostate cancer in Nature Reviews Clinical Oncology (2009) {Ref. 19}. Similarly, Alix-Panabières et al. discussed CTCs in breast, prostate, and colorectal cancer in the review article of Cancer Discovery (2016) {Ref. 6}. More recently, Fabisiewicz et al. reviewed CTCs in breast, colorectal, pancreatic, metastatic castration-resistant prostate cancer (mCRPC), and small cell lung cancer (SCLC) in the International Journal of Molecular Sciences (2024) {Ref. 7}.
While we acknowledge the reviewer’s point that combining multiple tumor types might risk misinterpretation of the scientific and clinical significance of CTCs, we believe that comparing CTC characteristics across colorectal and breast cancers may offer valuable insights into both commonalities and differences, which could benefit the readership.
Furthermore, we not only discuss the characterization of CTCs but also describe various CTC separation techniques in this review. Since these technologies are commonly applied across different types of cancer, including multiple cancer types in our discussion provides broader context and relevance for the readers.

Reviewer 2 Report
Comments and Suggestions for Authors
In this study, Kakizaki and colleagues focus their work on circulating tumor cells (CTCs) assays, heterogeneity, drug resistance, isolation technologies and ctDNA.
Overall, I found this review interesting; however, the manuscript must be yet improved and requires revisions.
Comments for improving the manuscript:
- The role of CTCs and ctDNA is not connected in the Introduction. This could be improved.
- Sub-heading “2. CTCs in the metastatic process” describes the spread of CTCs in a general way. Please include more information, such as apoptosis of CTCs in the circulation, possible interactions of CTCs with immune cells in the blood and the establishment of the pre-metastatic niche.
- I recommend the authors to include more about “survival versus death in the circulation of CTCs, and interaction with blood cells” (see also comment in point 2).
- It would be of advantage if the authors also could include DNA molecular profiling, not only transcriptomic profiling.
- In section 8.1, sub-heading 8. ”Future direction” is misleading, since 8.1 is more a summary of the previous sections.
- One of the challenges in 8.2 are defined as: “(1) Definition of CTCs using both gene expression and mutation profiles”. Indeed, the most common mutations for colorectal cancer are mentioned under sub-heading “ Heterogeneity and Molecular Pathology of CTCs”, but is not explored enough, especially regarding resistance induction.
Author Response
Reviewer 2:
Comments and Suggestions for Authors
In this study, Kakizaki and colleagues focus their work on circulating tumor cells (CTCs) assays, heterogeneity, drug resistance, isolation technologies and ctDNA.
Overall, I found this review interesting; however, the manuscript must be yet improved and requires revisions.
→ We thank Reviewer 2 for the time and effort to help improve our manuscript, and appreciate the encouraging comments. We have considered important points raised by Reviewer 2 and revised the manuscript as follows.
Comments for improving the manuscript:
1. The role of CTCs and ctDNA is not connected in the Introduction. This could be improved.
→ We thank the reviewer for the valuable comment. As suggested by Reviewer 2, we have added the following to the “1. Introduction” section.
“In addition to CTCs, another blood-based biomarker is circulating tumor DNA (ctDNA). ctDNA, released from apoptotic or necrotic tumor cells, serves as a biomarker for detecting somatic variants present in tumor tissue {Cohen, 2023}. Like CTCs, ctDNA can be obtained through a simple blood draw, offering a minimally invasive alternative to traditional biopsies. Although ctDNA cannot be used for transcriptional or pathological analyses, it has several clinical applications, including early cancer detection, detection of minimal/molecular residual disease (MRD), and monitoring of treatment response {Cohen, 2023}.” (P. 2 line 56)
2. Sub-heading “2. CTCs in the metastatic process” describes the spread of CTCs in a general way. Please include more information, such as apoptosis of CTCs in the circulation, possible interactions of CTCs with immune cells in the blood and the establishment of the pre-metastatic niche.
→ We thank the reviewer for the constructive comment. As suggested, we have added the following to the “2. CTCs in the metastatic process” section.
“Cancer cell–secreted factors, including tumor-derived extracellular vesicles and particles (EVPs), circulate and act on distant organs to establish a tumor microenvironment (TME) that is favorable for the initiation of metastasis. These changes lead to the formation of “pre-metastatic niches” (PMNs), which are receptive to the colonization of CTCs {Patras, 2023}. Tumor-associated macrophages (TAMs) are the particulary abundant stromal cells in TME of most solid tumors, and a high TAM density is associated with poor prognosis in cancer patients {Kitamura, 2015; Kitamura, 2023}.” (P. 2 line 88)
“Nevertheless, cancer metastasis is considered an inefficient process {Fidler, 1970}. CTCs are subjected to various forms of cell death due to factors such as detachment-induced apoptosis (anoikis), immune attacks by CD8⁺ T cells and natural killer (NK) cells {Kitamura, 2015}, and fluid shear stress as they traverse capillaries {Follain, 2020}.” (P. 3 line 96)
3. I recommend the authors to include more about “survival versus death in the circulation of CTCs, and interaction with blood cells” (see also comment in point 2).
→ We thank the reviewer for highlighting this important issue. As suggested by Reviewer 2, we have added the following to the “2. CTCs in the metastatic process” section.
“CD8⁺ T cells and NK cells typically induce apoptosis in CTCs through the secretion of perforin and granzymes, as well as through the expression of death ligands such as Fas ligand (FASL) and tumor necrosis factor–related apoptosis-inducing ligand (TRAIL) {Myers, 2021}. On the other hand, neutrophils contribute to both tumoricidal and tumor-promoting phenotypes {Fridlender, 2009}{Kitamura, 2015}{Szczerba, 2019}.
It is well established that CTCs undergoing epithelial-mesenchymal transition (EMT) express mesenchymal markers. Moreover, some CTCs have been reported to maintain a quiescent or dormant state {Nasr, 2023}. These observations suggest that the behavior of CTCs reflects not only their ability to adapt to the circulatory environment but also their inherent plasticity.“ (P. 3 line 99)
- It would be of advantage if the authors also could include DNA molecular profiling, not only transcriptomic profiling.
→ To address this issue, we attempted to analyze the available data on DNA molecular profiling of these CTCs. However, only limited information was available for six breast cancer CTCs, as shown in Auxiliary Figure 1. Since the amount of information is insufficient for a meaningful comparison with the ~300 transcriptomic profiles, we added the following note to the figure legend:
"As only limited genomic data on CTCs was available in these datasets, gene mutation data is not presented." (P. 5, line 182)
5. In section 8.1, sub-heading 8. ”Future direction” is misleading, since 8.1 is more a summary of the previous sections.
→ We thank the reviewer for highlighting this important point. As suggested by Reviewers 2 and 3, we have expanded the section previously titled “8. Future direction” and incorporated it into the new section “9. Precision Oncology Framework using CTCs,” which now begins on P. 10 line 368.
“After blood collection, CTCs are isolated using low-damage separation systems such as ClearCell FX1, CROSSORTER, or Parsortix (Sections 6 and 7). The isolated CTCs can then be processed using standard cytological techniques, including routine cytocentrifugation and immunocytochemistry. These cells can be examined and enumerated microscopically, or alternatively detected via signal amplification by qPCR (Section 5). CTC profiling provides valuable insights into the biology of metastasis and contributes to clinical decision-making, particularly in the context of molecularly targeted therapies (Sections 3 and 4).
CTCs may reflect metastatic potential, risk of recurrence, or prognostic relevance based on parameters such as cell count, the presence of gene mutations, or EMT characteristics (Sections 2 and 3). Even if CTCs are not initially detectable at the start of treatment, they may emerge during long-term monitoring, sometimes years later (Section 4). Furthermore, various immune cell populations can influence the survival and proliferation of CTCs or disseminated tumor cells (DTCs), which may enter reversible states of dormancy or quiescence (Section 3).
Thus, CTCs represent multifaceted biomarkers with significant promise in precision oncology, offering the potential to enhance diagnostic accuracy and enable individualized treatment strategies. A key advantage of CTC analysis lies in its minimally invasive nature, requiring only blood collection compared to conventional tissue biopsies.” (P. 10 line 368)
- One of the challenges in 8.2 are defined as: “(1) Definition of CTCs using both gene expression and mutation profiles”. Indeed, the most common mutations for colorectal cancer are mentioned under sub-heading “ Heterogeneity and Molecular Pathology of CTCs”, but is not explored enough, especially regarding resistance induction.
→ We thank the reviewer for the valuable comments. As suggested, we have added the following to the “3. Heterogeneity and Molecular Pathology of CTCs” section.
“In primary colorectal tumors, functional driver genes such as adenomatous polyposis coli (APC), tumor protein p53 (TP53), Kirsten rat sarcoma viral oncogene homolog (KRAS), and SMAD family member 4 (SMAD4) are associated with multistep tumorigenesis through the adenoma–carcinoma sequence {TCGA, 2012; Reiter, 2019} {Fearon, 1991; Kitamura, 2007}. Mutations in KRAS at codon 12 or 13 (observed in 35% to 45% of cases) or the BRAF V600E mutation (approximately 10%) confer resistance to cetuximab-based therapies in metastatic colorectal cancer (mCRC) {Bardelli, 2010}. Notably, KRAS and BRAF mutations are mutually exclusive in colorectal cancers {Rajagopalan, 2002}.” (P. 4 line 129)
“, suggesting a potential role in resistance to molecularly targeted therapies {Misale, 2012; Diaz, 2012}.” (P. 4 line 142)
“Such discrepancies may reflect tumor evolution under therapeutic pressure and the emergence of resistant clones {André, 2019; Robinson, 2013}.“ (P. 4 line 147)

Reviewer 3 Report
Comments and Suggestions for Authors
The authors of this manuscript describe circulating tumor cells and their possible use in patient therapy as prognostic markers. This manuscript is well written and well sectioned. The aim of the review is clearly stated and the arguments and conclusions appropriately described. There are a few suggestions that I state below that could improve the manuscript.
- The authors write “In contrast, CTCs can be analyzed if the appropriate equipment is available, and many detection systems are compatible with pathology laboratories to a certain degree.” in lines 200-201. Could the authors defend this posture a bit more since they talk about feasibility in this section. Especially since section 6 explains the separation techniques that don’t seem easily accessible to many hospitals or are not part of normal hospital equipment.
-There is a bit of a disbalance between sections 8.1 and 8.2 since the problems seem to be written more strongly than the benefits. Could the authors reinforce the benefits a bit more so they are not lost in the end of the manuscript where the “take-home message” tends to be.
- The figures are not of a very good quality and need to be neater. Please improve if possible.
- Please correct spelling errors ( like in line 87)
Author Response
Reviewer 3
The authors of this manuscript describe circulating tumor cells and their possible use in patient therapy as prognostic markers. This manuscript is well written and well sectioned. The aim of the review is clearly stated and the arguments and conclusions appropriately described.
→ We thank Reviewer 3 for the time and effort to improve our manuscript, and appreciate the encouraging comments. In the revised manuscript, we have made our best effort to strengthen our data by responding to the reviewer’s helpful commnets.
There are a few suggestions that I state below that could improve the manuscript.
- The authors write “In contrast, CTCs can be analyzed if the appropriate equipment is available, and many detection systems are compatible with pathology laboratories to a certain degree.” in lines 200-201. Could the authors defend this posture a bit more since they talk about feasibility in this section. Especially since section 6 explains the separation techniques that don’t seem easily accessible to many hospitals or are not part of normal hospital equipment.
→ To address this issue, we have added the following to the “5. Feasibility of the CTC test: A Comparison with the ctDNA Test” section.
“Once CTCs are isolated using non-destructive separation systems such as ClearCell FX1, CROSSORTER, or Parsortix (see Section 7) {Ciccioli, 2024; Oshiro, 2022; Warkiani, 2016}, they can be processed using standard cytological procedures. Specifically, the cells are attached to glass slides via routine cytocentrifugation, stained using conventional immunocytochemistry, and examined under a standard light microscope. For enhanced sensitivity, quantitative polymerase chain reaction (qPCR) can also be performed using Real-Time PCR instruments that are widely available in clinical laboratories (Table 1) {Yang, 2017}.” (P. 6, line 244)
-There is a bit of a disbalance between sections 8.1 and 8.2 since the problems seem to be written more strongly than the benefits. Could the authors reinforce the benefits a bit more so they are not lost in the end of the manuscript where the “take-home message” tends to be.
→ To address this issue, we have added the following. As suggested by Reviewers 2 and 3, we have expanded the section previously titled “8. Future direction” and incorporated it into the new section “9. Precision Oncology Framework using CTCs,” which now begins on P. 10 line 368.
“After blood collection, CTCs are isolated using low-damage separation systems such as ClearCell FX1, CROSSORTER, or Parsortix (Sections 6 and 7). The isolated CTCs can then be processed using standard cytological techniques, including routine cytocentrifugation and immunocytochemistry. These cells can be examined and enumerated microscopically, or alternatively detected via signal amplification by qPCR (Section 5). CTC profiling provides valuable insights into the biology of metastasis and contributes to clinical decision-making, particularly in the context of molecularly targeted therapies (Sections 3 and 4).
CTCs may reflect metastatic potential, risk of recurrence, or prognostic relevance based on parameters such as cell count, the presence of gene mutations, or EMT characteristics (Sections 2 and 3). Even if CTCs are not initially detectable at the start of treatment, they may emerge during long-term monitoring, sometimes years later (Section 4). Furthermore, various immune cell populations can influence the survival and proliferation of CTCs or disseminated tumor cells (DTCs), which may enter reversible states of dormancy or quiescence (Section 3).
Thus, CTCs represent multifaceted biomarkers with significant promise in precision oncology, offering the potential to enhance diagnostic accuracy and enable individualized treatment strategies. A key advantage of CTC analysis lies in its minimally invasive nature, requiring only blood collection compared to conventional tissue biopsies. CTC-based assays are expected to be integrated into clinical blood panels as tumor markers with greater clinical utility than traditional markers such as CEA, CA19-9, or CA15-3, thereby supporting oncologists in therapeutic decision-making.
Currently, over 3,000 clinical trials are investigating CTC-based diagnostics—approximately twice the number of trials focusing on circulating tumor DNA (ctDNA) [127]. These ongoing studies have demonstrated the feasibility and clinical relevance of CTC testing in prospective trials involving patients with colorectal and breast cancers [128-131].” (P. 6, line 368)
- The figures are not of a very good quality and need to be neater. Please improve if possible.
→ We thank Reviewer 3 for pointing out the issue of image quality in the PDF version. We have prepared and uploaded revised TIFF figures in a more compatible, high-resolution format. In addition to the Word file, we have also submitted a PDF version of the manuscript to ensure that the figures are displayed clearly. We hope that the updated figures now meet the required quality standards.
- Please correct spelling errors ( like in line 87)
→ We thank Reviewer 3 for pointing out the spelling errors, including the one on line 87. These have been corrected in the revised manuscript.
